# Cytotoxic, Antimicrobial, Antioxidant Properties and Effects on Cell Migration of Phenolic Compounds of Selected Transylvanian Medicinal Plants

**DOI:** 10.3390/antiox9020166

**Published:** 2020-02-18

**Authors:** Rita Csepregi, Viktória Temesfői, Sourav Das, Ágnes Alberti, Csenge Anna Tóth, Róbert Herczeg, Nóra Papp, Tamás Kőszegi

**Affiliations:** 1Department of Laboratory Medicine, University of Pécs, Medical School, Ifjúság u. 13, 7624 Pécs, Hungary; ritacsepregi93@gmail.com (R.C.); vtemesfoi@gmail.com (V.T.); pharma.souravdas@gmail.com (S.D.); 2János Szentágothai Research Center, University of Pécs, Ifjúság u. 20, 7624 Pécs, Hungary; herczeg.robert@pte.hu; 3Department of Pharmacognosy, Semmelweis University, Üllői út 26, 1085 Budapest, Hungary; albertiagnes@gmail.com (Á.A.); csenge512@gmail.com (C.A.T.); 4Department of Pharmacognosy, University of Pécs, Faculty of Pharmacy, Rókus u. 2, 7624 Pécs, Hungary; nora4595@gamma.ttk.pte.hu

**Keywords:** *Anthyllis vulneraria*, *Fuchsia magellanica*, *Fuchsia triphylla*, *Lysimachia nummularia*, antimicrobial activity, antioxidant capacity, fibroblasts, keratinocytes, cytotoxicity, cell migration

## Abstract

Medicinal plants are widely used in folk medicine but quite often their composition and biological effects are hardly known. Our study aimed to analyze the composition, cytotoxicity, antimicrobial, antioxidant activity and cellular migration effects of *Anthyllis vulneraria*, *Fuchsia magellanica*, *Fuchsia triphylla* and *Lysimachia nummularia* used in the Romanian ethnomedicine for wounds. Liquid chromatography with mass spectrometry (LC-MS/MS) was used to analyze 50% (*v/v*) ethanolic and aqueous extracts of the plants’ leaves. Antimicrobial activities were estimated with a standard microdilution method. The antioxidant properties were evaluated by validated chemical cell-free and biological cell-based assays. Cytotoxic effects were performed on mouse fibroblasts and human keratinocytes with a plate reader-based method assessing intracellular adenosine triphosphate (ATP), nucleic acid and protein contents and also by a flow cytometer-based assay detecting apoptotic–necrotic cell populations. Cell migration to cover cell-free areas was visualized by time-lapse phase-contrast microscopy using standard culture inserts. *Fuchsia* species showed the strongest cytotoxicity and the highest antioxidant and antimicrobial activity. However, their ethanolic extracts facilitated cell migration, most probably due to their various phenolic acid, flavonoid and anthocyanin derivatives. Our data might serve as a basis for further animal experiments to explore the complex action of *Fuchsia* species in wound healing assays.

## 1. Introduction

Nowadays, the investigation of natural extracts from medicinal plants has been increased due to their rich content of bioactive compounds such as polyphenols, vitamins and proteins, which are found in different parts of plants [1]. Phenolic compounds (flavonoids, phenolic acids, anthocyanins, tannins) are secondary metabolites, which play a crucial role in the pharmaceutical sciences, thanks to their extensive biological effects (antimicrobial, antioxidant, anticancer properties). These bioactive substances have diverse basic structures but possess an aromatic ring bearing one or more hydroxyl groups, which can be related to different biological impacts [2,3].

Medicinal plants are of primary importance in several regions of Transylvania, part of Romania. In our work, four plants were selected according to the previously described ethnomedicinal and phytochemical data. These medicinal plants are widely used in Transylvania, part of Romania. Another reason to select these herbs was that there are only a few scientific records in databases on them. The selected species are applied on wounds in traditional remedies in the country. *Anthyllis vulneraria* L. (common kidney-vetch, Leguminosae) is an annual, biennial or perennial plant that occurs in fields and meadows throughout Europe [4]. In Transylvania, the aerial part is used as an antiemetic drug [5], for swelling [6], wounds [7], kidney problems and diabetes as a tea [8] and as fodder [9]. In Lueta (local name: szipókavirág), it is used for wounds and stomach disorders as a tea (Nóra Papp, unpublished data). Its aerial part contains several bioactive compounds such as flavonoids, saponins [10,11], carotenoids, tannins and phenolic acids [12]. *Lysimachia nummularia* L. (creeping Jenny, Primulaceae) is an evergreen plant which lives mostly in ditches and wet grasslands, and in some places as a cultivated species throughout Europe [4]. In the Transylvanian ethnomedicine (local name: fillérfű, fillérlapi) the aerial part is used for toothache as a decoction [6], rheumatoid arthritis [13,14], wound and abscess as a fomentation [7,9,15,16], and pain of the legs as a fomentation and bath [17]. The leaves are rich in flavonoids [18] and triterpenoid saponins [19]. *Fuchsia magellanica* Lam. and *Fuchsia triphylla* L. (hardy fuchsia and lady’s eardrops, Onagraceae) are perennial cultivated plants all over in Europe, in addition, *F. magellanica* is locally naturalized, e.g., in Azores, Ireland, and Britain [4]. Fresh leaves of several *Fuchsia* varieties are ethnomedicinally applied on wounds [9], furuncles and skin inflammation as a fomentation [17,20]. Anthocyanins were detected in the flowers and berries of *Fuchsia* species [21,22], while in their leaves several flavonoids were present [23].

In our experiments, to get a comprehensive picture regarding the phenolic content, reversed-phase liquid chromatography coupled with tandem mass spectrometry (RP-LC-DAD-MS/MS) was used to tentatively characterize flavonoid and phenolic acid compounds. The main effects of the active constituents of the leaf extracts of selected plants in favor of wound healing are protection against microbial infection from the external environment, scavenging of free radicals with antioxidant effects and enhancing of cell migration, proliferation, angiogenesis and collagen production in the wounded area [24,25,26]. Therefore, the aim of the study was to test the antioxidant activity of the leaf extracts, where conventional total antioxidant capacity (TAC) chemical tests and cell-based antioxidant methods were applied. Besides that, we evaluated the antimicrobial properties by determination of the inhibitory effect of the leaf extracts against several Gram-positive and Gram-negative bacterial strains. The wound healing process means the interplay between various cell types, including neutrophils, macrophages, keratinocytes, fibroblasts, and endothelial cells [26,27]. For this reason, we applied fibroblast- and keratinocyte-based cellular models. To determine the nontoxic concentrations of the tested leaf extracts, we combined our previously published plate reader-based cell viability assay [28] with more sensitive flow cytometer-based fluorescence apoptotic–necrotic cell detection. Additionally, cell-based methods were performed by a time-lapse live imaging technique in order to measure the effects of the leaf extracts on the closure rate of standardized cell-free areas in a migration assay. These techniques enabled us to evaluate the biochemical properties of our plant extracts and to show their diverse biological effects on human keratinocyte and mouse fibroblast cell lines.

## 2. Materials and Methods

### 2.1. Reagents and Chemicals

Luminol (3-aminophthalhydrazide), 4-iodophenol, Trolox (6-hydroxy-2,5,7,8-tetramethylchroman-2-carboxylic acid), horseradish peroxidase (POD), Na_2_-fluorescein, AAPH (2,2′-azo-bis(2-amidinopropane) dihydrochloride), 2,2-Diphenyl-1-picrylhydrazyl (DPPH), potassium persulfate (K_2_S_2_O_8_), 2,2′-azino-bis(3-ethylbenzothiazoline-6-sulfonic acid) (ABTS), 2′,7′-dichlorofluorescein diacetate (DCFH-DA), dihydrorhodamine 123 (DHR123), quercetin, modified RPMI 1640 (supplemented with 165 mM MOPS, 100 mM glucose and 0.185 mM adenine), erythromycin, Dulbecco’s Modified Eagle Medium (DMEM), trypsin-EDTA, penicillin–streptomycin for cell culture, acetic acid and methanol of HPLC supergradient grade for LC-MS analyses, propidium iodide (PI), fluorescamine (Fluram) and 7-aminoactinomycin D (7AAD) were purchased from Sigma-Aldrich/Merck (Darmstadt, Germany). 3-(*N*-morpholino) propanesulfonic acid (MOPS) was from Serva Electrophoresis GmbH (Heidelberg, Germany). Ethanol (96% *v/v*, spectroscopic grade), glucose, adenine, agar-agar, and hydrogen peroxide (H_2_O_2_) were from Reanal Labor (Budapest, Hungary), while bioluminescent ATP Assay Kit CLSII and peroxide-free Triton X 100 (TX-100) were from Roche (Mannheim, Germany). Fetal bovine serum (FBS; Pan-Biotech, Aidenbach, Germany), and bovine serum albumin (BSA; Biosera, Nuaille, France) were used. Recombinant human platelet-derived growth factor-BB (PDGF-BB), phosphate-buffered saline (PBS, pH 7.4), Hanks’ Balanced Salt Solution (5.5 mM glucose) and Annexin V were from Thermo Fischer Scientific (Waltham, Massachusetts, USA). Highly purified water (<1.0 µS) was applied throughout the experiments. Plastic cell culture flasks and culture plates (96-well, 24-well and 6-well) were from TPP (Trasadingen, Switzerland), while standard 96-well plates were from Greiner Bio-One (Kremsmunster, Austria). For luminescence studies white 96-well optiplates were used (Perkin Elmer, Waltham, MA, USA).

### 2.2. Studied Plant Taxa and Plant Extraction

Voucher specimens of the selected four plants with unique codes were deposited at the Department of Pharmacognosy, University of Pécs, Pécs, Hungary. Fresh leaves of *Anthyllis vulneraria* (*A. vulneraria*, Voucher code: TR_7) and *Lysimachia nummularia* (*L. nummularia*, Voucher code: TR_15) were collected locally in Transylvania in July 2018, while *Fuchsia magellanica* (*F. magellanica*, Voucher code: TR_10) and *Fuchsia triphylla* leaves (*F. triphylla*, Voucher code: TR_9) were collected in June 2018 from the Botanical Garden of the University of Pécs, Pécs, Hungary.

Plant samples were dried at room temperature and stored in the dark. The plant extraction was performed according to the method of Lee et al. with some modifications [29].The aqueous and ethanolic extracts were obtained by extracting 3 g of leaf powder in 30 mL of 50% (*v/v*) ethanol or distilled water on an orbital shaker (Dual-Action Shaker KL2, Edmund Bühler GmbH, Bodelshausen, Germany) at room temperature overnight (200 rpm). The extracts were filtered through a 0.45 μm pore-size syringeless filter (Whatman Mini-UniPrep, Maidstone, United Kingdom), and further concentrated using a rotary vacuum evaporator (Rotavapor R-3, Buchi Labortechnik AG, Flawil, Switzerland). Amounts between 70–80 mg were obtained, which were dissolved in 1 mL of 50% (*v/v*) ethanol or distilled water. All prepared aqueous and ethanolic extracts were stored at −20 °C until the experiments were performed.

### 2.3. Analyses of Phenolic Compounds by HPLC with Diode-array Detector and Electrospray Ionization with MS

#### 2.3.1. HPLC Conditions

The chromatographic separation was performed on an Agilent 1100 HPLC system equipped with a G1379A degasser, G1312A binary gradient pump, G1329A autosampler, G1316A column thermostat and G1315C diode array detector (DAD) (Agilent Technologies, Waldbronn, Germany). Samples were separated on a Zorbax SB-C18 (Agilent Technologies, Santa Clara, CA, USA) (150 mm length, 3.0 mm i.d., 3.5 μm particle diameter) column, maintained at 25 °C. The mobile phase was composed of 0.3% acetic acid in water (*v/v*) (A) and methanol (B). The following gradient program was applied, at a flow rate of 0.3 mL/min with the composition of the mobile phase changing from 5% B to 100% B in 30 min, maintaining 100% B for 5 min and returning to 5% B in 1 min. All aqueous solvents were filtered through MF-Millipore (Millipore, Billerica, MA, USA) (0.45 μm, mixed cellulose esters) membrane filters. Chromatograms were acquired at 280 nm. Injection volume was 5 μL. Prior to injection, all samples were filtered through Sartorius (Goettingen, Germany) Minisart RC15 (0.2 μm) syringe filters.

#### 2.3.2. MS Conditions

Mass spectrometric analyses were performed with an Agilent 6410B triple quadrupole equipped with an electrospray ionization source (ESI) (Agilent Technologies, Palo Alto, CA, USA). ESI conditions were as follows: temperature: 350 °C, nebulizer pressure: 40 psi, N_2_ drying gas flow rate: 9 L/min, fragmentor voltage: 120 V, capillary voltage: 4000 V, collision energy was changed between 10 eV and 45 eV, depending on the analyzed structure. High purity nitrogen was used as collision gas. Full mass scan spectra were recorded in negative and for anthocyanins in positive ionization mode over the range of *m/z* 50–1000 Da (scan/s). The MassHunter B.01.03 software was used for data acquisition and qualitative analysis.

### 2.4. Determination of Minimum Inhibitory Concentration (MIC_80_) with Microdilution Method

All the bacterial strains were collected from Szeged Microbiology Collection (SZMC), Department of Microbiology, University of Szeged, Hungary, and from Pécs Microbiology Collection (PMC), Department of General and Environmental Microbiology, Institute of Biology, University of Pécs, Hungary. Tested strains were the followings: *Bacillus subtilis* (*B. subtilis*, SZMC strain: 0209), *Escherichia coli* (*E. coli*, PMC strain: 201), *Staphylococcus aureus* (*S. aureus*, ATCC strain: 29213), *Streptococcus pyogenes* (*S. pyogenes*, SZMC strain: 0119) and *Pseudomonas aeruginosa* (*P. aeruginosa*, PMC strain: 103). 

The microdilution method was performed according to a previously published method with some modifications [30]. In brief, 100 µL of bacterial suspensions (10^5^ CFU/mL) in modified RPMI 1640 and 100 µL of diluted aqueous or 50% (*v/v*) ethanolic leaf extracts in modified RPMI 1640 media were pipetted into each well of sterile 96-well plates. The sterile medium was considered as negative control, the inoculated RPMI 1640 without any treatment was taken as the bacterial growth control, while erythromycin was used as positive control. The final concentration of the ethanolic solvent for the dilution was restricted up to 1.0% v/v in the wells. The absorbance was measured at 595 nm on Multiskan EX 355 (Thermo Electron Corporation, Waltham, Massachusetts, USA) spectrophotometer, after 24 h incubation time at 30 °C. Absorbance values lower than 20% of the bacterial growth controls were considered as MIC_80_. Treatments were carried out with three technical replicates in five independent experiments.

### 2.5. Total Antioxidant Capacity (TAC) Assays

#### 2.5.1. Oxygen Radical Absorbance Capacity (ORAC) Assay

The ORAC test was executed according to the method of Kőszegi et al. without modifications [31]. The method is based on fluorescence quenching of Na_2_-fluorescein oxidized by AAPH. The quenching is delayed by the antioxidants present in the standards/samples. Serial dilutions of Trolox were used as standard. Briefly, into each well of normal 96-well plates 150 µL of working fluorescein solution (400 nM dissolved in 75 mM potassium phosphate buffer, pH 7.5) and 25 µL of blank/standard/plant extract (aqueous/ethanolic) were pipetted and the plates were preincubated for 30 min at 37 °C in the dark. After automated injection of 25 µL of AAPH solution (400 mM dissolved in 75 mM potassium phosphate buffer, pH 7.5) the fluorescence intensities were measured in kinetic mode for 80 min at 37 °C, with excitation and emission wavelengths of 490 and 520 nm, respectively. For the fluorescence measurements, the plate reader (BioTek Synergy HT, Winooski, Vermont, USA) was thermostated at 37 °C. Five independent experiments were done with three technical replicates for each treatment.

#### 2.5.2. Enhanced Chemiluminescence (ECL) Assay

The enhanced chemiluminescence method was performed following our previously published study without modifications [31]. The technique is based on the development of enhanced chemiluminescence (ECL) of luminol in the presence of peroxidase (POD), H_2_O_2_ and 4-iodophenol enhancer. The increase of the ECL signal is delayed, depending on the antioxidant capacity of the samples. Briefly, 70 µL of ECL detection reagent (0.15 M boric acid/NaOH, pH 9.6, supplemented with 0.45 mM luminol and 1.8 mM 4-iodophenol) and 200 µL POD enzyme solution (15 µU/mL) were premixed and kept on ice. Trolox dilutions were used as standard. Into each well of white optical 96-well plates 20 µL Trolox/blank/sample and 270 µL of POD-ECL reagent were added. The reaction was initiated by automated injection of 20 µL ice-cold H_2_O_2_ (1.5 mM, in 0.1% citric acid). The chemiluminescence signal was followed for 10 min, using a plate reader (Biotek Synergy HT) in kinetic analysis mode. Five independent experiments were done with three technical replicates for each treatment.

#### 2.5.3. 2,2-Diphenyl-1-Picrylhydrazyl (DPPH) Radical Scavenging Assay

The method is based on the absorbance decrease of DPPH, which is a stable organic radical. The measurement was conducted following the protocol described elsewhere [32,33], with some modifications. Briefly, 50 µL of blank/standard/plant sample dilutions followed by 100 µL of 200 µM DPPH (dissolved in 96% ethanol) and 50 µL of acetate buffer (100 mM, pH 5.5) were pipetted into 96-well general microplates. The absorbance changes were measured at 517 nm by a Perkin Elmer EnSpire Multimode plate reader (Perkin Elmer, Waltham, MA, USA) after 60 min incubation in the dark, at room temperature. The results were compared to serial dilutions of Trolox standard solution. Five independent experiments were done with three technical replicates for each treatment.

#### 2.5.4. Trolox Equivalent Antioxidant Capacity (TEAC) Assay

The technique is based on the generation of ABTS radical cation (ABTS•^+^) through the reaction between ABTS and potassium persulfate (K_2_S_2_O_8_). The method of Re et al. and Stratil et al. was adapted for the TEAC test with slight modification [34,35]. ABTS•^+^ was produced by reaction of ABTS stock solution (7 mM of ABTS dissolved in distilled water) with 2.45 mM K_2_S_2_O_8_ (final concentration) and diluted with PBS (pH 7.4) until the absorbance was 0.70 ± 0.005 at 734 nm. Then 20 μL aliquots of varying concentrations of the leaf extracts (50% ethanolic/aqueous) were allowed to react with 80 μL of ABTS•^+^ (7 mM) and the absorbance readings were recorded at 734 nm by the Perkin Elmer EnSpire Multimode plate reader after 20 min incubation in the dark, at room temperature. Trolox was used as standard. All measurements were carried out in five independent experiments with three technical replicates. 

#### 2.5.5. Calculation of Total Antioxidant Capacities (TAC)

For both the ORAC and ECL assays, the results were calculated as Trolox equivalents (TE). In the ORAC method the area under the fluorescence curve (AUC) of the blank was subtracted from that of the standard/sample (netAUC) and a calibration line was calculated for the netAUC of the Trolox standards. In the luminescence technique (ECL) the AUC of the emission curves vs. Trolox standards were used to calculate the calibration line. In both cases the samples’ TE values were obtained from the calibration curves which were then multiplied by the dilution factor and expressed as μM TE concentration. Finally, TAC was referred to 1 g of initial dry material for each plant sample. 

For the DPPH and TEAC assays, the radical scavenging activity was expressed as IC_50_ (the concentration of the plant extract in µg/mL, required to scavenge 50% of DPPH or ABTS reactions), calculated by a linear regression curve made from the scavenging activities vs. amount of extracts of the samples. This means that the lower the IC_50_ value of the sample is, the higher antioxidant activity it possesses. 

Radical scavenging activity of the leaf extracts in % of the blank was obtained using the following formula:(1)$$\mathrm{Radical}\;\mathrm{scavenging}\;\mathrm{activity}\;\left(\%\;\mathrm{inhibition}\right)=\;\left(\frac{A_{0}-A_{1}}{A_{0}}\right)\times 100$$
where A_0_ is the absorbance of the blank and A_1_ is the absorbance of the sample.

### 2.6. Cell Cultures

Mouse fibroblasts (3T3, ATCC: CRL-1658) were cultured in DMEM with high glucose (4500 mg/L), supplemented with 5% non-essential amino acids, 10% FBS, penicillin (100 U/mL) and streptomycin (100 µg/mL), while the human epidermal immortalized keratinocyte cell line (HaCaT) was kindly provided by the laboratory of Prof. Tamás Bíró (Department of Immunology, Faculty of Medicine, University of Debrecen, Hungary). HaCaT cells were cultured in DMEM with high glucose (4500 mg/L), supplemented with 10% FBS, penicillin (100 U/mL) and streptomycin (100 µg/mL) in 75 cm^2^ cell culture flasks at 37 °C in a humidified atmosphere containing 5% CO_2_. In both cases, after reaching 80% confluency, the cells were trypsinized and plated in 96-well/24-well/6-well sterile plastic plates.

### 2.7. Quantification of Intracellular ROS

Cellular oxidative stress due to the overproduction of reactive oxygen species (ROS) generated by AAPH was measured using the DCFH-DA and the DHR123 methods [36,37,38]. Trolox and quercetin were used as positive controls. The optimal conditions of DCFH-DA and DHR123 assays were as follows on 96-well culture plates: seeding density of 5 × 10^4^ cells/mL, cell culture incubation time for overnight. After washing with PBS co-incubation in Hanks’ (5.5 mM glucose) with 50 µM DCFH-DA or 10 µM DHR123 and plant extract/quercetin/Trolox on 3T3/HaCaT cell cultures for 1 h was performed. After removal of the treating medium and washing with PBS, 1 mM AAPH oxidant in Hanks’ glucose was added. The fluorescence intensity was recorded for 60 min on the Biotek microplate reader at 490/520 nm exc/em wavelengths at 37 °C. The radical scavenging activity was expressed as IC_50_ (the concentration of the plant sample (µg/mL), required to scavenge 50% of DCFH or DHR123 fluorescence), calculated by a linear regression analysis of the serial dilutions of the leaf extracts.

The radical scavenging activity was obtained using the following equation: (2)$$\mathrm{Radical}\;\mathrm{scavenging}\;\mathrm{activity}\;\left(\%\;\mathrm{inhibition}\right)\;=\;\left(\frac{{\mathrm{AUC}}_{0}-{\mathrm{AUC}}_{1}}{{\mathrm{AUC}}_{0}}\right)\times 100$$
where AUC_0_ is the area under curve values of the blank and AUC_1_ is the area under curve values of the sample. Five independent experiments were done with four technical replicates for each treatment.

### 2.8. Plate Reader Cytotoxicity Test

A multiparametric viability assay with one-step extraction was carried out following our previously published study without modifications to investigate the potential toxicity of 50% (*v/v*) ethanolic and aqueous leaf extracts [28]. *A. vulneraria* in 500–2500 µg/mL concentrations (ethanolic extracts) and 4000–8000 µg/mL concentrations (aqueous extracts) were tested. *F. magellanica* and *F. triphylla* in 50–800 µg/mL concentrations (ethanolic extracts) and 120–1000 µg/mL concentrations (aqueous extracts) were examined. *L. nummularia* in 250–1500 µg/mL concentrations (ethanolic extracts) and 3000–7000 µg/mL concentrations (aqueous extracts) were investigated. The final concentration of the ethanolic solvent was restricted up to 1.5% *v/v* in the wells, which concentration does not affect the viability of the cells. Briefly, 3T3 and HaCaT cells were treated with various concentrations of the plant extracts for 24 h, after that ATP was measured from the cell lysates with the bioluminescence method. Nucleic acid content was analyzed with PI staining, while intracellular proteins were quantified after fluorescent derivatization with fluorescamine. All results were expressed as mean ± SD in percentage compared with data obtained for the controls (~100%). Five independent experiments were done with four technical replicates for each treatment and dose response curves were calculated from the measured data. The dose-response curves were obtained after DoseResp fitting by using the OriginLab Pro software (version 2016, OriginLab Corporation, Northampton, MA, USA).

### 2.9. Flow Cytometric Cytotoxicity Test

In the plate reader analysis, we could estimate the cytotoxicity in general, however, using flow cytometry it is possible to reveal the type of potential cell injury induced by the leaf extracts. For this sensitive method we used lower concentrations of the 50% (*v/v*) ethanolic extracts; *A. vulneraria* in 50, 100, 200 µg/mL, *F. magellanica* and *F. triphylla* in 2.5, 5, 10 µg/mL and *L. nummularia* in 10, 25, 50 µg/mL concentrations, which were presumably sub-cytotoxic based on the plate reader viability assay. Experiments were carried out in three technical replicates on a BD Canto II cytometer (Becton, Dickinson and Company, Franklin Lakes, NJ, USA). Apoptotic, necrotic and late apoptotic cells were measured using Annexin V and 7-aminoactinomycin D (7AAD) staining. Annexin V was conjugated with fluorescein isothiocyanate (FITC) or Pacific Blue (PB), 7AAD was measured on the PerCP channel. Annexin V single positivity marked the apoptotic cells, 7AAD labeled the necrotic cells, the double positive population meant the late apoptotic cells, while the double negative population was live. Since two labels were used in one sample, positivity was defined based on fluorescence minus one (FMO) controls—which were the single-stained ones. Compensation and analysis was carried out in FlowJo v.10 (FlowJo LLC., Ashland, OR, USA). 

### 2.10. In Vitro “Wound Healing” Assay

The in vitro migration test was evaluated using culture inserts of 500 µm width (Ibidi GmbH, Gräfelfing, Germany). Briefly, an insert with 2 wells was placed in 24-well sterile culture plates, and then keratinocytes and fibroblasts were seeded into the 2 wells of the culture insert. After cell attachment and production of a monolayer, the culture insert was removed, and cells were incubated for 24 h with different sub-cytotoxic doses of 50% (*v/v*) ethanolic extracts. *A. vulneraria* in 50, 100, 200 µg/mL concentrations, *F. magellanica* and *F. triphylla* in 2.5, 5, 10 µg/mL concentrations and *L. nummularia* in 10, 25, 50 µg/mL concentrations were investigated. PDGF-BB was used as positive control at 15 ng/mL concentration. Within the cell-free gap the cell migration was visualized at every 4 h for 24 h by time-lapse imaging in bright field, using phase-contrast microscopy (JuLi Stage Real-Time Cell History Recorder, NanoEnTek, Seoul, Korea). The gap was monitored at an objective magnification of 10×. The closure rate of the open cell-free area was determined by quantifying the micro photo density data obtained for every occasion from the very same loci with ImageJ 1.x processing software (https://imagej.nih.gov/ij/). The closure rate in % was calculated by the following formula: (3)$$\mathrm{Closure}\;\mathrm{rate}\left(\%\right)=\left(\frac{{\mathrm{Open}\;\mathrm{area}}_{0.h}-{\mathrm{Open}\;\mathrm{area}}_{x.h}}{{\mathrm{Open}\;\mathrm{area}}_{0.h}}\right)\times 100$$
where “Open area _0.h_” is the cell-free area at the beginning of the experiment while “Open area _x.h_” is the still cell-free space at time points of imaging the samples. Finally, closure rate curves were constructed and the area under curve (AUC) for each treatment and cell line was calculated. The corresponding AUC data were averaged (±SD) and the summarized closure rates for the leaf extracts/PDGF were given in percentage (%) of the untreated controls. Three independent experiments were done with three technical replicates for each treatment. 

### 2.11. Statistical Analyses

Where appropriate, data were expressed in % of the control samples, which were assumed to be ~100%. In the cytotoxicity plate reader assay correlation coefficients of each tested parameter were given for the dose-response curves. Statistical evaluation was carried out in the antioxidant assays using independent t-test, where the ethanolic and aqueous extracts were compared with each other, furthermore, in the migration assay using one-way ANOVA test, where the control and the sample data of one type of treatment were compared by SPSS software (IBM, SPSS Statistics, version 22, Armonk, NY, USA). In addition, principal component analysis (PCA) was also used to test the differences between the two types of extracts of four selected medicinal plants and four chemical antioxidant assays. For PCA, prcomp function was used from the stats package within R (R Core Team 2019, version 3.6.1, Vienna, Austria). In all cases, the level of significance was set at *p* < 0.05. 

## 3. Results

### 3.1. Qualitative Analysis of Phenolic Compounds in Plant Extracts with LC-DAD-ESI-MS/MS 

Aqueous and 50% (*v/v*) ethanolic extracts of *A. vulneraria*, *F. magellanica*, *F. triphylla* and *L. nummularia* were studied using LC-DAD-ESI-MS/MS methods, in order to characterize the constituents responsible for the biological actions. Eighty-two gallic acid derivatives, hydroxycinnamic acid derivatives and flavonoid glycosides were detected altogether in the samples; moreover, eight anthocyanins were described in *Fuchsia* samples. Compounds were tentatively characterized by comparing their chromatographic behaviors, UV spectra and mass spectrometric fragmentation patterns with data from the literature. In order to provide semi-quantitative results regarding the quantities of the constituents, their relative abundance (%) was calculated according to the summarized areas of all the compounds that were detected in the UV (280 nm) chromatogram of the sample. Results are presented in Table 1 and Table 2, UV chromatograms (280 nm) of the extracts are shown in Appendix A).

Flavonol glycosides prevailed in *A. vulneraria* samples with compounds bearing the aglycone moieties kaempferol (e.g., compounds **23** and **25**) and quercetin (e.g., **40**), as well as methoxylated aglycone moieties isorhamnetin (e.g., **33**) and rhamnocitrin (e.g., **77**). In addition, the aqueous extract comprised diverse caffeoyl (e.g., **5**, **8**), coumaroyl (e.g., **7**, **10**) and feruloyl acid derivatives (e.g., **20**, **22**).

*L. nummularia* also contained flavonol glycosides in high amounts, however, the samples were dominated by the presence of myricetin glycosides (e.g., **43**, **45**, **68**, **69**) with myricetin 3-*O*-desoxyhexoside (**49**) as the main compound detected in both the aqueous and 50% (*v/v*) ethanolic extracts, and other aglycones were hardly found. For the aqueous samples, occurrence of coumaroyl (**7**, **14**, **15**) and feruloyl glucarate (**18**, **20**, **22**) isomers was characteristic. 

Unusual quercetin galloyl hexosides (**46**, **50**, **53**) and a kaempferol galloyl hexoside (**58**) were detected in *Fuchsia* samples and primarily in *F. magellanica.* Even if anthocyanins were predominant, a peonidin dihexoside isomer (**V**) was present in each *Fuchsia* sample, while cyanidin dihexoside (**II**) was described only in *F. magellanica* extracts.

The compound analyses data are summarized in dendrograms for the ethanolic and aqueous leaf extracts separately in Appendix A.

### 3.2. Determination of Minimum Inhibitory Concentration (MIC_80_)

The effects of leaf extracts on Gram-positive and Gram-negative bacteria were determined by CLSI M07-A9 (Vol. 32, No. 2) guidelines (Table 3). For this study, Minimum Inhibitory Concentration (MIC_80_) values under 100 µg/mL were considered to indicate good antimicrobial activity; from 500 to 100 µg/mL to show moderate antimicrobial activity; from 1000 to 500 µg/mL to point to weak antimicrobial activity; and over 1000 µg/mL to indicate inactivity. 

The ethanolic extracts of *Fuchsia* species were considered to have good antibacterial activities (MIC_80_ values were between 5 and 60 µg/mL concentrations) on *S. aureus*, *B. subtilis, S. pyogenes* and *P. aeruginosa*, compared with erythromycin, where the MIC_80_ values were between 0.1–42 µg/mL concentrations. Ethanolic extracts of *A. vulneraria* and *L. nummularia* showed moderate antimicrobial activity on *B. subtilis* and *S. pyogenes*, and *A. vulneraia* pointed to weak antimicrobial activity on *S. aureus*. 

Interestingly, only the aqueous extracts of *Fuchsia* spp. of the tested plants indicated a remarkably good influence on *S. aureus*, *B. subtilis* and *S. pyogenes (*MIC_80_ values between 17 and 65 µg/mL concentrations). The plant extracts did not have an antibacterial effect on *E. coli*.

### 3.3. Total Antioxidant Capacity (TAC) Assays

The antioxidant properties of ethanolic and aqueous extracts of the investigated plants were evaluated by conventional chemical assays such as DPPH, TEAC, ORAC, and ECL methods (Figure 1). The lowest IC_50_ values indicate the highest antioxidant effect with DPPH and TEAC scavenging activity tests, while the highest TE/g values mean the strongest antioxidant capacity in case of ORAC and ECL assays. Generally, the ethanolic extracts showed higher antioxidant effect than aqueous extracts. Altogether, the ethanolic and aqueous extracts of *Fuchsia* species had the strongest antioxidant activity followed by *Lysimachia nummularia* and *Anthyllis vulneraria* in all methods.

The results of Principal Component Analysis (PCA) are shown in Figure 2. In the case of ethanolic extracts, a strong positive correlation was observed between the ORAC and ECL methods, while a moderate correlation existed between the data of TEAC and DPPH assays. These relationships were opposite to the results of aqueous extracts because of their stronger correlation between TEAC and DPPH methods than ECL and ORAC tests. The ethanolic extracts of *Fuchsia* species were more similar to each other than in the case of their aqueous extracts. 

### 3.4. Inhibition of Intracellular ROS Production

The oxidation of DCFH and DHR was generated by peroxyl radicals from AAPH in 3T3 and in HaCaT cells [57]. The leaf extracts decreased the fluorescence of DCF and the calculated 50% inhibition values are shown in Figure 3. Ethanolic and aqueous extracts of *Fuchsia* species had the strongest inhibition in both cell lines, compared with other plant extracts. However, no antioxidant activity was detected for *A. vulneraria* in the case of 3T3 cells, and only ethanolic extracts of *A. vulneraria* showed measurable antioxidant property on HaCaT cell culture.

The generation of fluorescence signal from the rhodamine derivative was also decreased by the plant extracts and the calculated 50% inhibition values are shown in Figure 4. Both solvent fractions of *Fuchsia* spp. exerted the highest inhibition of the fluorescence intensity of rhodamine. Although ethanolic and aqueous extracts of *A. vulneraria* could be quantified, the aqueous fraction had only weak effectivity (at around the detection limit).

### 3.5. Plate Reader Cytotoxicity Tests 

Cytotoxicity data obtained for fibroblast (3T3) and keratinocyte cell cultures (HaCaT) are seen in Figure 5. The ethanolic extracts reduced the ATP, cell number and protein contents of both cell lines in a dose dependent manner more effectively than the aqueous extracts. Nevertheless, these differences between the type of solvents were not observed in case of *Fuchsia* species. Both ethanolic and aqueous extracts of *Fuchsia*s had the same toxic effects in the investigated cell lines. In 3T3 cells, the decrease of ATP, cell number and protein levels were more pronounced than in HaCaT cultures.

Evaluating the results of compound analyses, antimicrobial, antioxidant and cytotoxicity data we decided to use only the ethanolic extracts in the further experiments because of their richer ingredient content and stronger biological effects. Potentially sub-toxic concentrations of the ethanolic extracts (*A. vulneraria* in 50, 100, 200 µg/mL, *F. magellanica* and *F. triphylla* in 2.5, 5, 10 µg/mL and *L. nummularia* in 10, 25, 50 µg/mL concentrations) were used in all further investigations. 

### 3.6. Flow Cytometric Cytotoxicity Test

Our apoptosis–necrosis results for the hypothesized nontoxic concentrations of ethanolic extracts are summarized for 3T3 cells in Table 4 and for HaCaT cells in Table 5. The two different cell lines gave similar results, namely the applied concentrations of the leaf extracts did not cause significant apoptosis/necrosis and most of the cells remained intact (the number of dead/dying cells was under 4% in both cell lines). An example of the gating strategy can be seen in the Appendix A section (Appendix A).

### 3.7. In vitro Migration Test

To investigate the impact of the ethanolic leaf extracts on the migration of fibroblasts and keratinocytes a standardized 500 µm cell-free area was created in 24-well culture plates by Teflon inserts. Figure 6 shows the effects of *A. vulneraria* and *L. nummularia* extracts while Figure 7 those of *F. magellanica* and *F. triphylla* on the migration of 3T3 and HaCaT cells. 

The positive control PDGF-BB at 15 ng/mL concentration had a strong significant stimulating effect on cell migration (117.05 ± 6.72% on 3T3 and 115.16 ± 8.26% on HaCaT cells), compared with untreated control cells. Of the tested four plants, *A. vulneraria* expressed a slight stimulatory effect (107.01 ± 7.35% in 100 µg/mL concentration and 104.54 ± 8.86% in 200 µg/mL concentration) only on HaCaT cells, while enhanced migration and closure rate were observed in *F. magellanica* and *F. triphylla* treated cells when compared with untreated cells. Moreover, these extracts reached the response of the positive control, PDGF. Of the two *Fuchsia* species, *F. magellanica* had the strongest incentive effects on both cell lines, because the closure rate was 120.26 ± 10.17% on 3T3 cells and 114.61 ± 3.72% on HaCaT cells, respectively, in 2.5 µg/mL concentration. *L. nummularia* did not show any significant migration effect towards the two cell lines.

On the other hand, it is noteworthy that the less concentrated leaf extracts produced the most efficient stimulation on cell migration when compared with extracts of higher concentrations.

## 4. Discussion 

Our investigated medicinal plants are used in Transylvanian folk medicine for the treatment of various skin diseases. Of the tested plants, we could not find scientific studies to evaluate their biological effects, especially on cell migration and proliferation. Although we have some preliminary results of antioxidant effects and phytochemical data of the extracts of *Anthyllis vulneraria* and *Fuchsia* species [58], to our best knowledge, the present study is the first to analyze the cytotoxicity and combined effects of the ethanolic extracts in a cellular migration model that mimics wound healing ability.

Bioactive agents of plants may have specific functions on wound healing properties, including antioxidant, antimicrobial, anti-inflammatory effects via migration, proliferation and pro-collagen stimulating actions and they can modulate one or more phases of the wound healing process. For instance, tannins and flavonoids have anti-inflammatory and antibiotic effects [59,60]. Flavonoids and phenolic acids are well-known antioxidant compounds, and the position and degree of hydroxylation are essential to their activity. In flavonoids, *O*-dihydroxy structure on the B ring, 2,3 double bond in conjugation with a 4-oxofunction on the C ring, the presence of a 3-hydroxyl group on C ring and C3- and C5-OH moieties in a combination with 4-oxofunction on the A and C rings are required for maximum radical scavenging potential [2,61]. In phenolic acids, the presence of a 3-hydroxyl structure (2,3-hydroxybenzoic acid, gallic acid, caffeic acid and caftaric acid) enhances the antioxidant effect [2,62]. Anthocyanins are strong antioxidant compounds and their scavenging activity corresponds with their structural characteristics as well, di-acylated forms possess higher antioxidant activity than the mono-acylated and non-acylated anthocyanins [63]. Sun et al. demonstrated that peonidin and cyanidin-based anthocyanins exert strong antioxidant properties. Furthermore, they verified, that acylated forms have high antibacterial effect [64]. On the other hand, anthocyanins from various berries have a strong anti-inflammatory effect via inhibition of IL-1β, Il-6 and COX-2 gene expression. Moreover, these colorful compounds stimulate the wound healing with increasing the migration of fibroblasts and keratinocytes [65,66]. 

We found that in *A. vulneraria* the most abundant amount of flavonol compounds are in glycosidic or methoxylated forms with lower activities than the aglycones [2]. In contrast, *Fuchsia* species contained, besides common flavonol compounds, various cinnamic acid and benzoic acid derivatives such as caffeic acid, ellagic acid and gallic acid derivatives, kaempferol– and quercetin–galloyl–glycosides. Moreover, the leaves of *F. magellanica* and *F. triphylla* contained several anthocyanins, such as cyanidin and peonidin derivatives. In the leaf extracts of *L. nummularia*, the main flavonoids were myricetin derivatives, which have strong antibacterial effects; however, in spite of this observation we detected weaker activities, compared to the standard antibiotic control and also to *Fuchsia* species [67]. The main differences between the investigated plants are that *Fuchsia* species contain high amounts of anthocyanins and hydroxybenzoic acids, which are well-known antioxidants and antimicrobial constituents; moreover, it has been proved, that they have a stimulating effect on cell migration [2,24,67]. 

Nevertheless, there are some contradictions in the literature regarding cell proliferation and migration effects of phenolic compounds, which act through different signaling pathways to express their effect in various cell lines. The polyphenols of tea suppress the migration of proliferation abilities of tumor cells by inhibiting NF-κB activation and quenching the expression of cyclin D1 [68]. The other compound carnosol inhibits the migration and tumor growth via ROS-dependent proteasome degradation of STAT3 in several breast cancer cell lines [69]. LaFoya et al. investigated that some flavonoids, including resveratrol, apigenin, chrysin, genistein, luteolin, and myricetin, significantly inhibited the endothelial cell migration and proliferation in a statistically significant manner with the moderation of Notch signaling, while quercetin did not regulate these processes [70]. Several studies demonstrated that phenolic acids, such as caffeic acid, coumaric acid and ferulic acid, which belong to hydroxycinnamic acids, have anti-proliferative and apoptotic effects on tumor cells [71,72,73,74]. On the other hand, the caffeic acid phenethyl ester can stimulate wound re-epithelization and enhance the proliferation of keratinocytes [75]. Ly et al. have proved that chlorogenic acid has migration and invasion stimulating effects on trophoblasts through an adenosine monophosphate-activated protein (AMP) kinase-dependent pathway, while not affecting cell proliferation [76]. However, the hydroxybenzoic acids, such as ellagic acid and gallic acid, increase the proliferation and migration of keratinocytes and fibroblasts in a dose-dependent manner [77]. 

During the past decades, a wide variety of analytical methods has been developed for the measurement of the total antioxidant capacity [78]. Several studies found good linear correlation between the total phenolic and/or flavonoid content and antioxidant capacities of plants. In agreement with other researchers, we found correlations between the TEAC and DPPH assay as well as for ORAC and ECL methods [31,79,80,81]. On the other hand, these differences between antioxidant methods can be explained by their chemical backgrounds. The underlying mechanism of ECL and ORAC assays is the hydrogen atom transfer (HAT), while TEAC and DPPH tests are based on single electron transfer (SET) reaction [81]. The assessment of total antioxidant capacity by cell-free chemical assays cannot completely reflect the behavior of the complex plant samples in vivo. Therefore, it is important to evaluate the effectiveness of antioxidants in more biologically relevant conditions, such as testing of the compounds in cell-based antioxidant assays [82]. The two used fluorogenic reporter molecules are sensitive to peroxyl radicals with similar oxidative mechanisms, therefore, DHR and DCFH gave closely matching scavenging activities as was expected from literature data [57]. 

In wound healing assays it is important to differentiate between proliferation and migration because the cells are usually not synchronized. Therefore, some authors use mitomycin C DNA synthesis inhibitor pretreatment to follow the migration solely when extended (48 h) incubation time is applied because it is longer than the generation cycle of the cells [83]. In our tests, we did not apply this compound because our incubation time was only 24 h, and the proliferation ability is minimal within this short period. HaCaT cell culture is an immortalized human keratinocyte cell line, with a doubling time of approximately 24 h (36.2 ± 1.5 h in the early passages (2–8) and 24 ± 0.6 h in the late passages (12–16) [84], while 3T3 mouse fibroblast cells have a doubling rate of 20–26 h [85].

In our migration tests, nontoxic doses of each extract were applied, where the fibroblast and keratinocyte cells showed more than 90% cell viability by the plate reader assay, and the necrotic (dead) cells were under 4% in the flow cytometric assay. 3T3 fibroblast cell culture was more vulnerable to exposure of the leaf extracts than HaCaT keratinocytes. One possible explanation of the increased sensitivity to potentially cytotoxic plant compounds might be that fibroblasts with the epidermal phenotype are located in the dermis, while keratinocytes are mostly found in the basement membrane zone (epidermis); therefore they should be more resistant against external noxae [86]. Our phytochemical and biological data suggest that *Fuchsia* species exerted the highest positive effect on cell migration among the four tested plants.

The found antimicrobial IC_50_ values of *Fuchsia* species were lower in the antibacterial assay than IC_50_ cytotoxicity values found for the fibroblasts and keratinocytes. Therefore, one may conclude that *Fuchsia* species extracts may possess effective components for the formulation of new antibacterial and wound healing stimulatory agents. 

In our planned further experiments, another powerful approach for the characterization of the mode of action of the test compounds (i.e., polyphenols mediating the pathways of wound healing via modulation of various biomarkers such as growth factors, matrix metalloproteinases and cytokines) could be the measurement of secreted factors from the supernatant or the analysis of intracellular signaling pathways by Western blot and molecular biological methods. These studies are inevitably necessary for identifying the impact of complex plant extracts in wound healing processes [87,88].

## Figures and Tables

**Figure 1 antioxidants-09-00166-f001:**
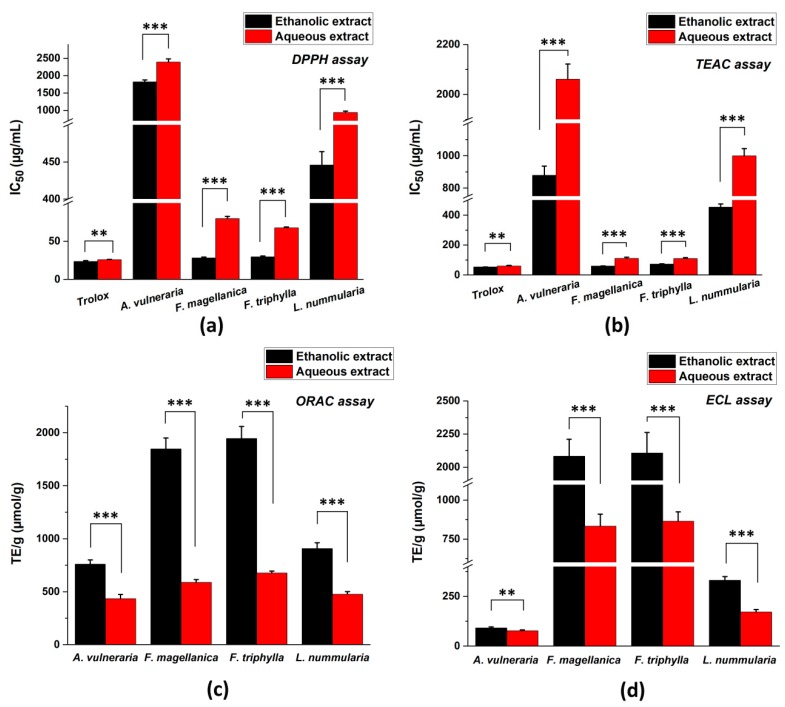
Total antioxidant capacity (TAC) of four selected medicinal plants measured by different spectroscopic methods: (**a**) DPPH assay; (**b**) TEAC assay; (**c**) ORAC assay and (**d**) ECL assay. IC_50_ values (in μg/mL concentration at 50% inhibition) were calculated in case of DPPH and TEAC methods, while TE/g values (Trolox equivalent in µmol referred to 1 g of initial dry material) were determined in case of ORAC and ECL tests. Mean ± SD of 5 independent experiments, each in 3 replicates. The aqueous and ethanolic extracts were compared with t-probe (** *p* < 0.01, *** *p* < 0.001).

**Figure 2 antioxidants-09-00166-f002:**
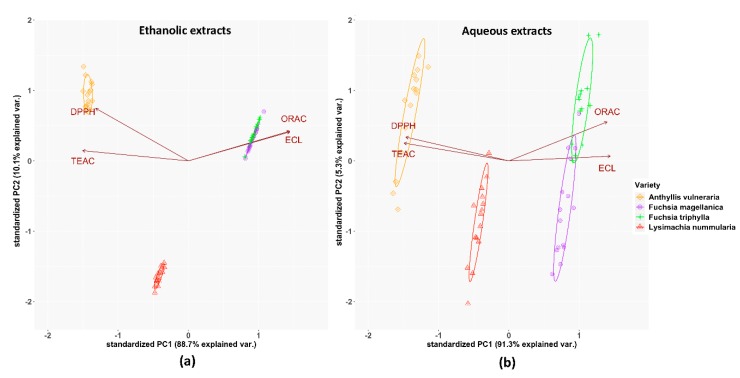
PCA analysis of the investigated plant species’ total antioxidant capacity by four independent methods: (**a**) in the ethanolic extracts, (**b**) in the aqueous solutions, based on the antioxidant activities. % of total variance explained by each axis is provided within the figure. The symbols indicate 15 individual data for each plant extract.

**Figure 3 antioxidants-09-00166-f003:**
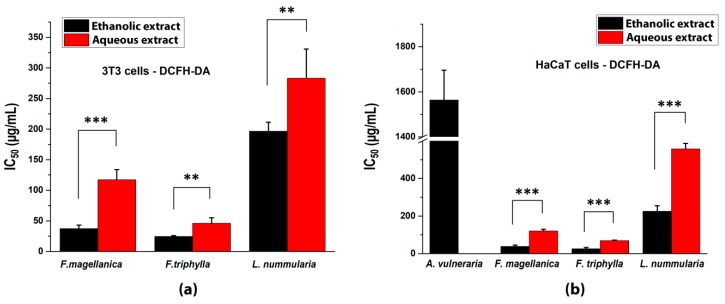
Intracellular antioxidant capacities of studied plant extracts with DCFH-DA staining (**a**) in 3T3 fibroblast cells and (**b**) in HaCaT keratinocyte cells. IC_50_ inhibitory concentrations were calculated from the equations obtained for the inhibitory capacities of the serial dilutions of the extracts. In the case of *A. vulneraria* the DCFH-DA scavenging activity data (when not shown) were at around/below the detection limit. Mean ± SD of five independent experiments, *n* = 5 × 4 replicates for each concentration. The aqueous and ethanolic extracts were compared with t-probe (** *p* < 0.01, *** *p* < 0.001).

**Figure 4 antioxidants-09-00166-f004:**
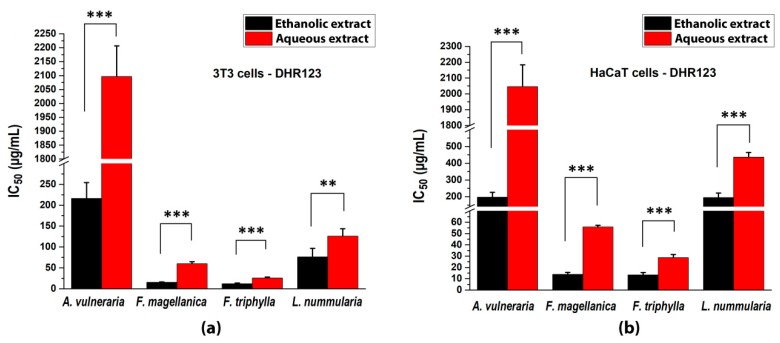
Intracellular antioxidant capacities of studied plant extracts with DHR123 (**a**) on 3T3 fibroblast cells and (**b**) on HaCaT cell culture. 50% inhibitory concentrations were calculated from the equations obtained for the inhibitory capacities of the serial dilutions of the extracts. Mean ± SD of 5 independent experiments, *n* = 5 × 4 replicates for each concentration. The aqueous and ethanolic extracts were compared with t-probe (** *p* < 0.01, *** *p* < 0.001).

**Figure 5 antioxidants-09-00166-f005:**
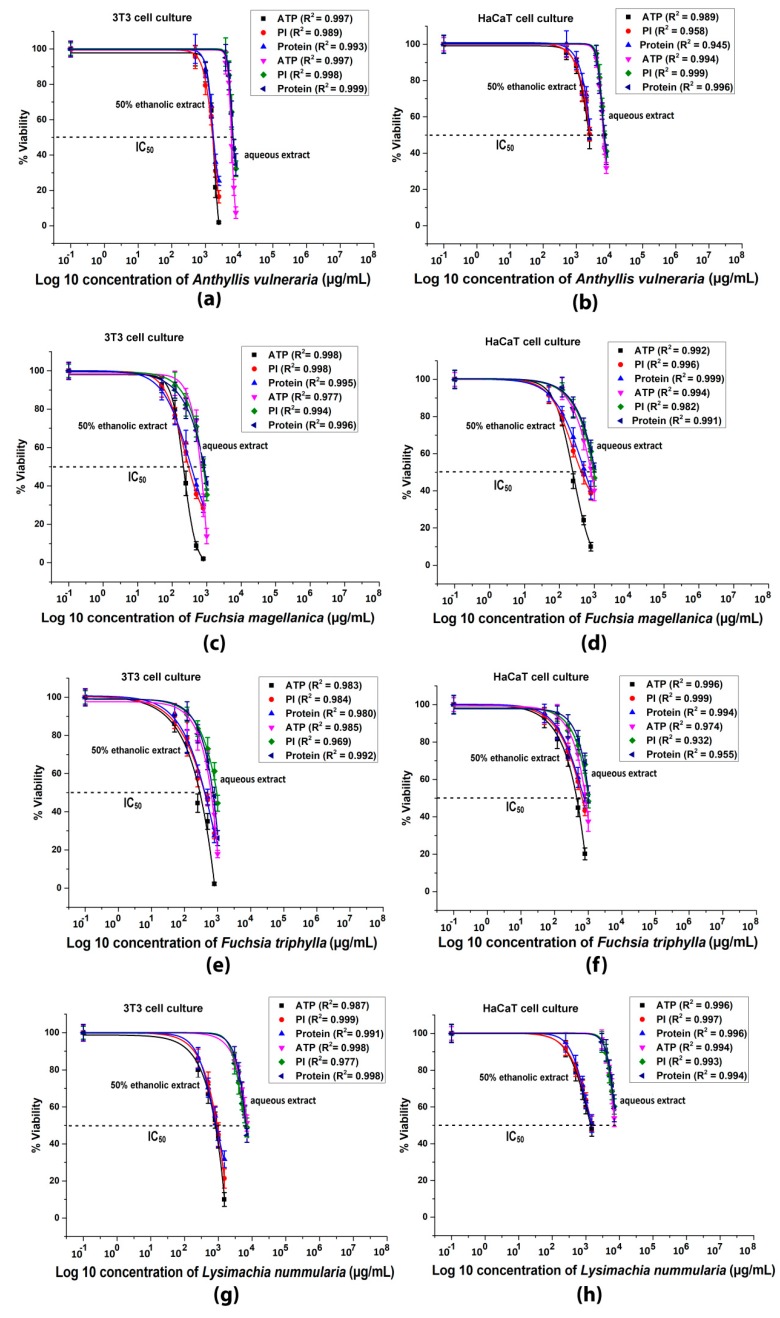
Intracellular ATP content, cell number (PI) and total protein levels of 3T3 and HaCaT cells with dose-response fitting: (**a**,**b**) cytotoxic effects of *A. vulnearia* on 3T3 and HaCaT cells; (**c, d**) cytotoxic effects of *F. magellanica* on 3T3 and HaCaT cells; (**e**,**f**) cytotoxic effects of *F. triphylla* on 3T3 and HaCaT cells; (**g**,**h**) cytotoxic effects of *L. nummularia* on 3T3 and HaCaT cells. Data are expressed in % of the control. Mean ± SD of 5 independent experiments, *n* = 5 × 8 replicates for each concentration. Dose-response curves were created by log10 transformation and nonlinear curve fitting. Correlation coefficients (*R^2^*) were calculated for each treatment.

**Figure 6 antioxidants-09-00166-f006:**
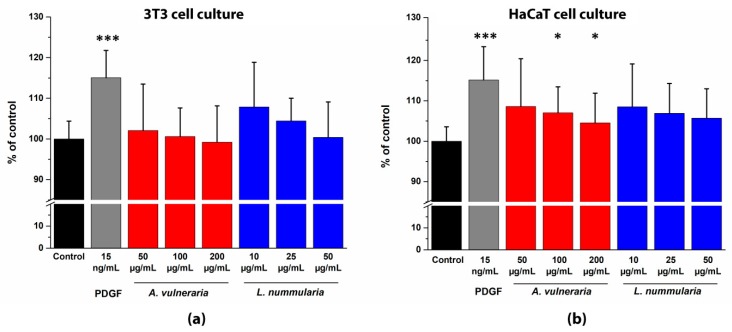
Comparative analysis by time-lapse imaging of wound closure ability of PDGF-BB and leaf extracts. (**a**) 50% (*v/v*) ethanolic extracts of *A. vulneraria* and *L. nummularia* on 3T3 monolayer cultures; (**b**) 50% (*v/v*) ethanolic extracts of *A. vulneraria* and *L. nummularia* on HaCaT monolayer cultures. Cells were monitored for 24 h in the absence and in the presence of plant extracts or PDGF with phase contrast microscopy. The bar graphs of AUC were calculated from cumulative closure rates (CR %) studied at every 4 h with three different concentrations from each plant. Mean ± SD of three independent experiments. *n* = 3 × 3 replicates. One-way Anova test (* *p* < 0.05, *** *p* < 0.001 compared with control).

**Figure 7 antioxidants-09-00166-f007:**
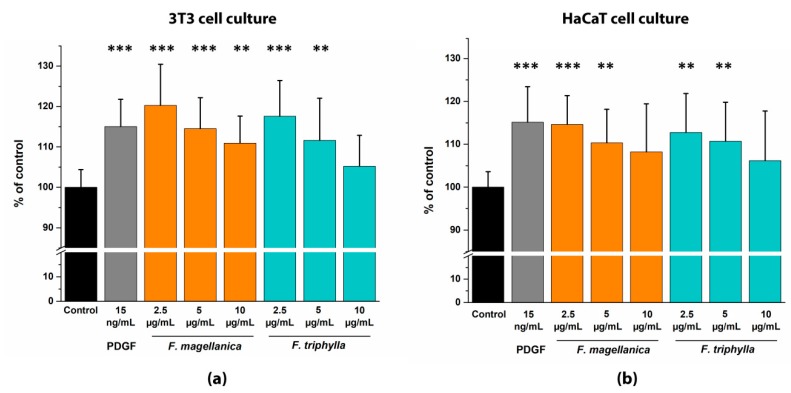
Comparative analysis by time-lapse imaging of wound closure ability of PDGF-BB and plant extracts on HaCaT monolayer. (**a**) 50% (*v/v*) ethanolic extracts of *F. magellanica* and *F. triphylla* on 3T3 monolayer cells; (**b**) 50% (*v/v*) ethanolic extracts of *F. magellanica* and *F. triphylla* on HaCaT monolayer cultures. Cells were monitored for 24 h in the absence and in the presence of plant extracts or PDGF with phase contrast microscopy. The bar graphs of AUC were calculated from cumulative closure rates (CR %) studied at every 4 h with three different concentrations from each plant. Mean ± SD of three independent experiments. *n* = 3 × 3 replicates. One-way Anova test (** *p* < 0.01, *** *p* < 0.001 compared with control).

**Table 1 antioxidants-09-00166-t001:** LC-MS/MS data and tentative characterization of compounds from *Anthyllis vulneraria, Fuchsia magellanica, Fuchsia triphylla* and *Lysimachia nummularia* leaf extracts.

**No.**	t_R_ (min)	λ_max_ (nm)	[M−H]^−^ (*m/z*)	Fragment ions (*m/z*)	Tentative Characterization ^a^	Presence and Relative Abundance (%) of Compounds in the Leaf Extracts ^b^								Ref.
						AvE^c^	AvW^c^	FmE^c^	FmW^c^	Ft E^c^	FtW^c^	LnE^c^	LnW^c^	
1	2.0	315	343	179, 135	Caffeic acid derivative	0.1								[39]
2	2.2	306	533, 375,	217, 173, 149	Cinnamoylquinic acid derivative		0.3							[39]
3	2.5	315	341 (683)	179, 149, 135	Caffeoyl-*O*-hexoside (dimer)	0.2		0.3					0.2	[40]
4	2.6	265, 312	337	267, 191, 163, 149, 135	5-*p*-Coumaroylquinic acid				0.2	0.1	0.1			[39]
5	3.2	288sh, 311	639, 353	191	Caffeoylquinic acid derivative	0.3	0.9							[39]
6	7.1	272	331	169, 125	Galloyl hexoside						0.2		0.2	[41]
7	9.5	311	355	191	Coumaroyl glucarate isomer		0.1						2.9	[42]
8	10.1	298sh, 320	371	209, 191, 179	Caffeoylquinic acid derivative		0.5							[39]
9	10.5	260	611	305	n.i.							2.0	1.8	-
10	10.5	298, 310	355	209, 191, 163	Coumaroyl glucarate isomer		0.5						1.3	[42]
11	10.6	298, 320	549	387, 369, 267, 249, 137	Cinnamic acid derivative		0.3							[43]
12	10.6	298	331	169, 125	Galloyl hexoside								2.6	[41]
13	10.9	300	301	168, 150, 125	Galloyl pentoside		0.2							[41]
14	11.9	282sh, 307	355 (711)	271, 209, 191	Coumaroyl glucarate isomer (dimer)								3.4	[42]
15	12.6	283sh, 312	355	271, 209, 191	Coumaroyl glucarate isomer		1.9					5.0	6.9	[42]
16	13.0	290, 328sh	297, (595)	179, 161, 135	Caffeic acid derivative (dimer)			2.0	2.4		4.0			[39]
17	13.0	296	575	413, 351, 267, 249, 163, 113	Coumaric acid derivative		0.5							[39]
18	13.1	310	385	209, 191	Feruloyl glucarate							2.2	3.9	[42]
19	13.2	255, 350	787	625, 462, 301, 299	Quercetin 3-*O*-hexosyl-hexosyl-7-*O*-hexoside		1.0							[44]
20	13.3	283sh, 312	385	209, 191	Feruloyl glucarate		0.7					3.0	10.2	[42]
21	13.5	265, 349	771	609, 462, 301, 299, 285, 284, 283, 179	Quercetin 3-*O*-hexosyl-desoxyhexosyl-7-*O*-hexoside	2.2	1.0							[44]
22	13.8	288sh, 312	385	271, 209, 191, 163, 146, 119	Feruloyl glucarate		3.0						2.3	[42]
23	13.9	265, 349	771	609, 446, 445, 285, 284, 283, 179	Kaempferol 3-*O*-hexosyl-hexosyl-7-*O*-hexoside	2.6	0.9							[44]
24	13.9	276	475, 453	-	n.i.						2.5			-
25	14.4	264, 351	755	593, 446, 284, 283	Kaempferol 3-*O*-hexosyl-desoxyhexosyl-7-*O*-hexoside	2.5	2.9							[45]
26	15.0	272	651	399, 325, 163	n.i.		1.9			2.9				-
27	15.2	256, 354	625	463, 462, 301, 299	Quercetin 3-*O*-hexosyl-7-*O*-hexoside	1.6	1.2							[45]
28	15.7	257, 352	595	462, 433, 301, 299, 271	Quercetin 3-*O*-pentosyl-7-*O*-hexoside	0.7	1.2							[41]
29	15.8	268	305	225, 147, 135	n.i.			0.8	0.5			0.9	1.3	-
30	16.2	265, 347	609	447, 446, 285, 283	Kaempferol 3-*O*-hexosyl-7-*O*-hexoside	1.7	0.5							[41]
31	16.4	278	449	357, 275	n.i.							1.5	0.9	-
32	16.5	288sh, 321	575	443, 267, 249, 193, 175	1,3-*O*-Diferuloylglycerol pentoside		0.7							[46]
33	16.6	254, 354	639	519, 477, 461, 315, 314, 313, 299, 151	Isorhamnetin 3-*O*-hexosyl-7-*O*-hexoside	4.6	1.4							[47]
34	17.2	268, 295sh	693	477, 345, 327, 315, 300, 207, 183, 165	n.i.						1.8			-
35	17.4	252, 269sh, 332	443	267, 249, 193, 175, 149, 134, 113	1,3-*O*-Diferuloylglycerol	5.1	4.4							[46]
36	17.5	252, 269sh, 352	963	801, 625	Flavonoid		0.6							[43]
37	18.0	260	197		n.i.					3.2				-
38	18.0	268, 350	639	319, 301, 283, 239, 213, 203, 197, 157, 142, 130, 116, 109	Flavonoid	6.2	6.5							[43]
39	18.1	276	521	337, 191, 163	Coumaroylquinic acid derivative							1.2	0.7	[39]
40	18.2	266, 350	625	463, 300, 271, 255, 243, 179	Quercetin 3-*O*-hexosyl-hexoside	2.9	2.5							[47]
41	19.2	274	387 (775)	169, 151, 124	Gallic acid derivative (dimer)			0.4	0.2					[46]
42	19.3	274	537	271, 211, 169, 151, 124	Gallic acid derivative			0.2	0.1					[46]
43	19.3	266, 357	625	479, 306	Myricetin 3-*O*-desoxyhexosyl-hexoside							2.3	1.5	[48]
44	19.3	266, 355	609	429, 285, 284, 255, 227	Kaempferol 3-*O*-hexosyl-hexoside	1.2	2.6							[41]
45	19.4	267, 357	479	316	Myricetin 3-*O*-hexoside							1.1	0.8	-
46	19.5	262, 355	615	463, 301, 300, 271, 169	Quercetin galloyl hexoside			0.4	0.4	1.8	1.0			[47]
47	19.6	264, 352	639	459, 315, 314, 257	Isorhamnetin 3-*O*-hexosyl-hexoside		1.9							[49]
48	19.8	266, 331	593	429, 284, 255, 227	Kaempferol 3-*O*-desoxyhexosyl-hexoside	2.7	1.7							[41]
49	20.2	260, 354	463	317, 316	Myricetin 3-*O*-desoxyhexoside							22.0	17.2	[50]
50	20.3	255, 293sh, 359	615	301	Quercetin galloyl hexoside			0.1	0.1					[47]
51	20.5	259, 356	477	301, 283, 255, 179, 151, 121	Quercetin glucuronide			0.5	0.2	0.6	0.6			[47]
52	20.6	257, 268sh, 356	433	301, 300	Quercetin 3-*O*-pentoside			0.1	0.1					[47]
53	20.7	257, 268sh, 356	615	301	Quercetin galloyl hexoside			0.9	0.5					[51]
54	20.7	260, 353	463	317, 316	Myricetin 3-*O*-desoxyhexoside							2.4		[50]
55	20.8	257, 355	463	301, 300, 271, 255, 243, 179, 163, 151	Quercetin 3-*O*-hexoside	3.6	4.9	0.1	0.1	0.6	0.5			[41]
56	20.9	266, 350	579	463, 315, 313	Isorhamnetin-3-*O*-pentosyl-7-*O*-pentoside	5.0	3.8							[47]
57	20.9	257, 355	609	463, 301, 300, 299	Quercetin 3-*O*-desoxyhexosyl-7-*O*-hexoside			0.1	0.1	0.2	0.5	0.8	1.8	[48]
58	21	257, 355	599	447, 313, 285, 169	Kaempferol galloyl hexoside			0.1	0.1					[43]
59	21.3	256, 356	433	300, 271, 255, 151	Quercetin 3-*O*-pentoside	2.4	2.0	0.2	0.1	0.4	0.4			[47]
60	21.5	254, 368	301	284, 245, 229, 201, 185, 145, 129, 117	Ellagic acid			0.3	0.1	1.1	0.6			[51]
61	21.7	266, 350	447	284, 255, 227, 151	Kaempferol 3-*O*-hexoside	2.6	2.2	0.2	0.1					[41]
62	21.7	266, 350	447	301, 300, 271, 255, 179, 151	Quercetin 3-*O*-desoxyhexoside					1.1	1.2			[47]
63	21.8	266, 332	705	437, 407, 325, 245, 231, 199, 163, 121	n.i.	2.4	1.8							-
64	21.8	266, 350	521	331, 271, 211, 169	Galloyl hexoside derivative			0.5	0.3					[41]
65	21.9		517	267, 249, 205, 161, 113	n.i.	2.0	1.2							-
66	22.0	266, 347	447	284, 255, 227	Kaempferol 3-*O*-hexoside			0.4	0.2					[41]
67	22.0	255, 350	477	315, 314, 285, 271, 257, 243	Isorhamnetin 3-*O*-hexoside	1.8	4.2							[52]
68	22.0	264, 340	477	331, 317	Myricetin desoxyhexoside derivative							2.8	3.4	[53]
69	22.0	264, 340	447	331, 317	Myricetin derivative							2.5	1.0	[53]
70	22.3	266, 340	639	477, 459, 315, 314, 267	Isorhamnetin 3-*O*-hexosyl-hexoside	1.1	2.6							[52]
71	22.3	266, 347	417	285, 284, 255, 227	Kaempferol 3-*O*-pentoside			0.3	0.2					[41]
72	22.4	267, 328	727	551, 491, 415, 267, 249	Ferulic acid derivative	2.0								[54]
73	22.4		623	431, 371, 345, 317, 301, 299	n.i.							0.8	0.5	-
74	22.5	267, 328	727	551, 415, 267, 183	Ferulic acid derivative		1.4							[54]
75	22.8	255, 266sh, 334	447	314, 285, 271, 257, 243	Isorhamnetin 3-*O*-pentoside		1.3							[52]
76	22.9	266, 351	599	447, 301, 300, 179, 151	Quercetin 3-*O*-desoxyhexoside derivative					0.1	0.1			[47]
77	23.4	266, 348	623	443, 299, 298, 283, 271	Rhamnocitrin 3-*O*-dihexozid	3.8	3.2							[54]
78	23.7	268, 339	609	315, 314, 193	Isorhamnetin 3-*O*-pentosyl-hexoside	0.6	1.5							[52]
79	23.8	268, 329	799	623, 485, 397, 299, 298	n.i.	2.1	0.7							-
80	24.8		477	301, 267, 249, 227, 209, 183, 165, 113	Ellagic acid derivative		0.6							[51]
81	24.9	250, 370	531	301, 300	Quercetin derivative					0.1	0.5			[47]
82	24.9	266, 344	593	413, 299, 298, 283	Rhamnocitrin 3-*O*-hexosyl-pentoside	1.1	0.7							[54]

^a^ Compound numbers and retention times (tR) refer to UV chromatograms shown in Appendix A (Appendix A). ^b^ Relative abundance: area% of the compound calculated from the summarized areas of all compounds detected in the UV (280 nm) chromatogram. ^c^ Abbreviations: AvE: *Anthyllis vulneraria* 50% (*v/v*) ethanolic extract, AvW: *Anthyllis vulneraria* aqueous extract, FmE: *Fuchsia magellanica* 50% (*v/v*) ethanolic extract, FmW: *Fuchsia magellanica* aqueous extract, FtE: *Fuchsia triphylla* ethanolic extract, FtW: *Fuchsia triphylla* aqueous extract, LnE: *Lysimachia nummularia* 50% (*v/v*) ethanolic extract, LnW: *Lysimachia nummularia* aqueous extract, n.i.: not identified.

**Table 2 antioxidants-09-00166-t002:** LC-MS/MS data and tentative characterization of anthocyanins from *Fuchsia magellanica* and *Fuchsia triphylla* leaf extracts.

No.	t_R_ (min)	[M + H]^+^ (*m/z*)	Fragment Ions (*m/z*)	Tentative Characterization ^a^	Presence and Relative Abundance (%) of Compounds in the Leaf Extracts ^b^				Ref.
					FmE^c^	FmW^c^	FtE^c^	FtW^c^	
I	2.0	767	453, 153	Anthocyanin (n.i.)	1.1				-
II	2.7	611	449, 287	Cyanidin dihexoside	3.0	18.3			[55,56]
III	2.9	783	303	Anthocyanin (n.i.)			5.9		-
IV	9.1	625	463, 301	Peonidin dihexoside	15.7				[55,56]
V	10.3	625	463, 301	Peonidin dihexoside	18.9	39.9	5.3	5.4	[55,56]
VI	10.7	487	325, 185	Anthocyanidin hexoside		4.2			-
VII	11.4	441	249	Anthocyanin (n.i.)				9.6	-
VIII	11.8	411	249	Anthocyanin (n.i.)			6.1	9.6	-

^a^ Compound numbers and retention times (t_R_) refer to UV chromatograms shown in Appendix A (Appendix A). ^b^ Relative abundance: area% of the compound calculated from the summarized areas of all compounds detected in the UV (280 nm) chromatogram. ^c^ Abbreviations: FmE: *Fuchsia magellanica* 50% (*v/v*) ethanolic extract, FmW: *Fuchsia magellanica* aqueous extract, FtE: *Fuchsia triphylla* 50% (*v/v*) ethanolic extract, FtW: *Fuchsia triphylla* aqueous extract.

**Table 3 antioxidants-09-00166-t003:** Minimum inhibitory concentration (MIC_80_) of selected medicinal plant extracts on *E. coli*, *P. aeruginosa*, *S. aureus*, *B. subtilis* and *S. pyogenes*.

		MIC_80_ (µg/mL)	
Test Bacteria	Treatment	Ethanolic Extracts	Aqueous Extracts
*Escherichia coli*	*erythromycin*	30.56–42.88	
	*A. vulneraria*	N.D.	N.D.
	*F. magellanica*	N.D.	N.D.
	*F. triphylla*	N.D.	N.D.
	*L. nummularia*	N.D.	N.D.
*Pseudomonas aeruginosa*	*erythromycin*	69.91–82.76	
	*A. vulneraria*	N.D.	N.D.
	*F. magellanica*	55.03–58.91	N.D.
	*F. triphylla*	49.03–58.87	N.D.
	*L. nummularia*	N.D.	N.D.
*Staphylococcus aureus*	*erythromycin*	0.09–0.25	
	*A. vulneraria*	715.90–835.92	N.D.
	*F. magellanica*	4.81–7.65	17.13–25.03
	*F. triphylla*	5.32–6.73	30.24–36.99
	*L. nummularia*	1063.75–1181.29	N.D.
*Bacillus subtilis*	*erythromycin*	0.10–0.18	
	*A. vulneraria*	288.02–313.01	N.D.
	*F. magellanica*	13.91–15.72	52.73–65.03
	*F. triphylla*	14.55–22.66	33.16–45.33
	*L. nummularia*	160.90–164.28	N.D.
*Streptococcus pyogenes*	*erythromycin*	0.08–0.09	
	*A. vulneraria*	220.06–236.30	N.D.
	*F. magellanica*	11.83–14.99	41.72–44.91
	*F. triphylla*	11.61–17.92	41.95–43.23
	*L. nummularia*	179.43–203.56	N.D.

Five independent experiments, each in three replicates, compared with erythromycin as positive control. N.D.: not detected.

**Table 4 antioxidants-09-00166-t004:** Apoptosis-necrosis assay data of 3T3 cells treated with 50% (*v/v*) ethanolic extracts of *A. vulneraria*, *F. magellanica*, *F. triphylla* and *L. nummularia* with Annexin V-7AAD flow cytometric method.

3T3 Cells				
Treatment Groups	Annexin V–7AAD Method			
	Live Cells (%)	Necrotic Cells (%)	Early Apoptotic Cells (%)	Late Apoptotic Cells (%)
Control cells	97.13	2.06	0.66	0.15
50 µg/mL *A. vulneraria*	96.29	2.46	1.00	0.25
100 µg/mL *A. vulneraria*	96.00	2.90	0.83	0.27
200 µg/mL *A. vulneraria*	96.89	2.22	0.70	0.20
2.5 µg/mL *F. magellanica*	96.95	2.03	0.75	0.26
5 µg/mL *F. magellanica*	96.27	2.67	0.79	0.27
10 µg/mL *F. magellanica*	96.62	2.25	0.78	0.34
2.5 µg/mL *F. triphylla*	97.74	1.26	0.84	0.17
5 µg/mL *F. triphylla*	98.15	1.01	0.71	0.13
10 µg/mL *F. triphylla*	97.57	1.50	0.62	0.30
10 µg/mL *L. nummularia*	95.63	2.98	1.05	0.33
25 µg/mL *L. nummularia*	95.33	3.31	1.07	0.29
50 µg/mL *L. nummularia*	95.48	2.81	1.34	0.38

**Table 5 antioxidants-09-00166-t005:** Apoptosis–necrosis assay data of HaCaT cells treated with 50% (*v/v*) ethanolic extracts of *A. vulneraria*, *F. magellanica*, *F. triphylla* and *L. nummularia* with Annexin V–7AAD flow cytometric method.

HaCaT Cells				
Treatment Groups	Annexin V–7AAD Method			
	Live Cells (%)	Necrotic Cells (%)	Early Apoptotic Cells (%)	Late Apoptotic Cells (%)
Control cells	99.04	1.31	0.49	0.47
50 µg/mL *A. vulneraria*	96.84	1.56	0.51	1.09
100 µg/mL *A. vulneraria*	95.29	3.36	0.51	0.83
200 µg/mL *A. vulneraria*	96.06	2.60	0.35	0.99
2.5 µg/mL *F. magellanica*	97.76	1.39	0.47	0.37
5 µg/mL *F. magellanica*	97.08	2.19	0.38	0.34
10 µg/mL *F. magellanica*	97.56	1.61	0.54	0.28
2.5 µg/mL *F. triphylla*	97.98	1.06	0.54	0.42
5 µg/mL *F. triphylla*	97.68	1.33	0.54	0.45
10 µg/mL *F. triphylla*	98.04	0.91	0.63	0.42
10 µg/mL *L. nummularia*	97.19	1.70	0.42	0.69
25 µg/mL *L. nummularia*	96.89	1.86	0.41	0.84
50 µg/mL *L. nummularia*	96.15	2.49	0.44	0.91

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
