# Peer review of "Cytotoxic, Antimicrobial, Antioxidant Properties and Effects on Cell Migration of Phenolic Compounds of Selected Transylvanian Medicinal Plants"

_antioxidants, 2020, doi:10.3390/antiox9020166_

Round 1

Reviewer 1 Report

The manuscript is interesting and within the aim and scope of Antioxidants. Authors choose four plants used in Transylvanian ethnomedicine with wound healing properties and tested their cytotoxic, antimicrobial and antioxidant activity.

Keywords: some of keywords are too general, e.g. plate reader, flow cytometry, cell migration. The name of plant should be added to keywords

Line 55: “fillerfu, fillerlapi” are  the local name of Transylvanian ethnomedicine?

Line 109-112: justification for the choice of plants used in investigation should be given in Introduction. Remove those lines.

Line 123: What was the final volume of the extracts after concentration.

Were HPLC conditions based on literature? Lack of references.

Gradient program is unclear, e.g. from 0 to 30 min. mobile phase composition was constant?

Line 182: Reedit the number of section.

Section 327-334: Authors mistakenly used the term “aglycones”, e.g. compounds 23 and 25 are not “aglycones kaempferol”; they are derivatives.

Reviewer 2 Report

The study was carefully designed and the experimental approach is consistent with the research field and the aims of the journal.

Manuscript deserves publication after these suggested corrections:

-Quality of figures should be improved (mandatory).

-Statistical significance values are often missing in the figure captions.

-A wide plethora of secondary metabolites were tentatively identified through HPLC-MS analysis. Did the author perform orthogonal analysis on selected compounds for more accurate quantitative determination?

-Finally, authors could further improve the manuscript value through a pharmacological investigation of a selected biomarker substantiating the wound healing process.

Reviewer 3 Report

The manuscript reports the polyphenolic composition, citooxicity, antimicrobial, antioxidant activity and cellular migration effects of selected Romanian (specifically Transylvanian) medicinal plants. Among the species studied, Fuchsia ones showed the strongest cytotoxicity and the highest antioxidant and antimicrobial activity. The work is interesting and can serve as a basis for further animal experiments to explore the complete action of Fuchsia species in wound healing assays.

The work is interesting and conducted with adequate means. However some remarks are needed to be clarified prior to acceptance.

Line 66. The authors claimed they managed either to “identify” or “tentatively identify” the polyphenolic content of their extracts. Unfortunately the term “identify” cannot be used (also elsewhere) since no reference standard was used for this purpose.  Section 2.2. Did the authors employ a validated approach? Regarding Table 1. The relative abundance reported in not exhaustive. No reference standard was used, but anyway I invite the authors to provide numeric quantification data.

Round 2

Reviewer 3 Report

The authors have substantially improved their manuscript and it can be now accepted for publication in the present form.